# Review of Orthogonal Frequency Division Multiplexing-Based Modulation Techniques for Light Fidelity

**Rahmayati Alindra** ID**, Purnomo Sidi Priambodo \*** ID **and Kalamullah Ramli** ID

Department of Electrical Engineering, Faculty of Engineering, Universitas Indonesia, Depok 16424, Indonesia; rahmayati.alindra@ui.ac.id (R.A.); kalamullah.ramli@ui.ac.id (K.R.)
\* Correspondence: purnomo.sidhi@ui.ac.id

**Abstract:** Light Fidelity (LiFi) technology has gained attention and is growing rapidly today. Utilizing light as a propagation medium allows LiFi to promise a wider bandwidth than existing Wireless Fidelity (WiFi) technology and enables the implementation of cellular technology to improve bandwidth utilization. In addition, LiFi is very attractive because it can utilize lighting facilities consisting of light-emitting diodes (LEDs). A LiFi system that uses intensity modulation and direct detection requires the signal of orthogonal frequency division multiplexing (OFDM) to have a real and non-negative value; therefore, certain adjustments must be made. The proposed methods for generating unipolar signals vary from adding a direct current, clipping the signal, superposing several unipolar signals, and hybrid methods as in DC-biased optical (DCO)-OFDM, asymmetrically clipped optical (ACO)-OFDM, layered ACO (LACO)-OFDM, and asymmetrically clipped DC-biased optical (ADO)-OFDM, respectively. In this paper, we review and compare various modulation techniques to support the implementation of LiFi systems using commercial LEDs. The main objective is to obtain a modulation technique with good energy efficiency, efficient spectrum utilization, and low computational complexity so that it is easy for us to apply it in experiments on a laboratory scale.

**Keywords:** light fidelity; optical orthogonal frequency division multiplexing; modulation techniques; light-emitting diodes

## 1. Introduction

The demand for broadband internet connectivity across the globe has progressively surged over the years, primarily driven by an escalating number of personal devices owned by consumers, the evolution of social media platforms, and the widespread adoption of the Internet of Things (IoT) technology [1–3]. Radio frequency (RF) technology with 300 GHz bandwidth is no longer sufficient to accommodate the increasing data traffic, the better solution is to provide transmission technology using a visible-light spectrum and infrared that has an aggregate bandwidth of about 780 THz [4]. Therefore, the term visible light communication (VLC) emerged; this system functions by utilizing a transmitter with an LED and a receiver with a photodiode. The transmission of data occurs through the modulation of LED light intensity, and the detection of the received signal is accomplished through the use of a photodiode employing the direct detection principle. VLC can overcome the problem of electromagnetic interference and high latency in RF technology [5]; moreover, it has been touted as one of the cutting-edge technologies to support the 5th generation (5G) wireless networks [6] and 6G technology [7]. The concept of VLC then extended into LiFi to attain a robust wireless communication that is characterized by high data transfer rates, bidirectional transmission, and full network integration while also ensuring the security of transmitted data [8].

The data rate of the LiFi is influenced by two factors, namely lighting technology and modulation techniques [4,9]. The light sources are in lighting technology, such as phosphor-coated blue LED, red-green-blue (RGB) LEDs, Gallium Nitride (GaN) micro LEDs, and RGB

lasers. The achievable data rates reported are 1 Gbps for phosphor-coated blue LED with an additional blue filter in the receiver [10], 5 Gbps for RGB LED [11], 8 Gbps for GaN micro-LED [12], and 100 Gbps for RGB lasers [13]. Different characteristics of the light sources affect the available bandwidth, and data rate can be achieved. Among the above-mentioned light sources, phosphor-coated blue LED is the most widely available on the market and commonly used as a means of lighting in various places. However, the phosphor material slows down the frequency response where the higher the frequency, the greater the attenuation, so the bandwidth that can be provided is only about 2 MHz to 5 MHz [14]. In addition to using a blue filter, a proposed technique to increase the modulation bandwidth of phosphor-coated blue LEDs includes multi-band OFDM modulation, where each OFDM signal band is allocated to a specific LED chip in an LED lamp [15].

The implementation of LiFi is challenged by the limited bandwidth of LEDs. To overcome this problem, the utilization of modulation techniques that exhibit high spectral efficiency is needed [16]. The LiFi modulation schemes can be categorized into four groups, namely single carrier modulation (SCM), OFDM-based multi-carrier modulation (MCM), other MCM, and color modulation [17]. SCM techniques, such as on-off keying (OOK), pulse amplitude modulation (PAM), pulse width modulation (PWM), and pulse position modulation (PPM), are relatively easy to implement, but they were suitable for low to medium data rates. Due to the limitation of LED bandwidth, SCM techniques will experience inter-symbol interference (ISI) at high bit rates, and a complex equalizer is needed to deal with this [18]. In contrast, MCM-OFDM techniques have immunity against ISI, and to handle high data rates, a single tap equalizer is sufficient. Hence, concerning power and spectrum efficiency, OFDM performs better than SCM [19,20]. Moreover, OFDM supports adaptive power and bit loading to increase performance and even supports multiuser communication so that it is massively used in optical wireless communication [21,22]. However, the intensity modulation on the LED requires that the transmitted signal is real and non-negative. Furthermore, the conventional OFDM must be refined to realize the intensity modulation (IM) and direct detection (DD) principle on the LED.

The third group of LiFi modulation techniques is the other multi-carrier modulation, which includes discrete Hartley transformation (DHT), wavelet packet division multiplexing (WPDM), and Hadamard coded modulation (HCM). Generally, these three techniques replace the Fast Fourier Transform (FFT) block in OFDM systems. DHT eliminates Hermitian symmetry, resulting in lower computational complexity compared to conventional OFDM based on FFT. Meanwhile, HCM was developed as a viable solution to fulfill dimming level requirements. Lastly, WPDM utilizes discrete Wavelet packet transform instead of FFT, yielding increased spectrum efficiency. Unfortunately, WPDM is not efficient in terms of power efficiency because it still requires additional DC bias to achieve unipolar values [23].

Subsequently, the last group is the color domain modulation, which is predicted to increase the degree of freedom of the LiFi system. Color shift keying (CSK), as proposed in the IEEE 802.15.7 standard as a modulation technique for VLC, uses RGB LED as the light source and CIE 1931 chromatic color space as a constellation map [24]. The main research on CSK modulation recently is optimizing the constellation design for the CSK format. Still, there is a lack of experimental information, especially for high-order constellations [25] and unclear receiver structure [17]. Therefore, for the sake of implementation, OFDM-based modulation is considered more affordable and more modest than color modulation. However, it does not rule out the possibility of combining the CSK method with OFDM. By utilizing two-level laser diode-based CSK OFDM, a data rate of 28.4 Gbps can be attained on the transmitter and receiver, which are 1.25 m apart [26].

It should be noted that most of the aforementioned modulation schemes are presented in the paper exposing the formulas, and then their performance depends on the simulation of the proposed algorithm. However, a more practical methodology is required to evaluate the algorithm's reliability and its opportunities to be realized in functional

hardware. Therefore some researchers have been exploring the use of Field Programmable Gate Arrays (FPGA) to design high-speed integrated systems [27–30]. FPGA offers advantages, including high flexibility, fast development times, and a low-cost solution to implement LiFi communication systems. Furthermore, the advanced signal processing algorithms can be synthesized in FPGA to improve error correction and increase the system's overall throughput.

In this paper, we review and compare various modulation techniques with a focus on OFDM-based techniques. This is to answer the need to implement a LiFi system using commercial phosphor-coated blue LEDs. The main objective is to obtain a modulation technique that has good energy efficiency, efficient spectrum utilization, and an OFDM variant that has low computational complexity so that it is easy to apply in experiments on a laboratory scale, including electronic circuitry and embedded systems.

The rest of this paper is organized as follows. Section 2 describes the numerous schemes of OFDM-based modulation with their merits and demerits. Section 3 describes four important performance criteria when selecting an OFDM technique. Section 4 begins with a comparison of OFDM techniques based on the performance criteria then the best-performing technique is reviewed in detail. Finally, conclusions are presented in Section 5.

## 2. OFDM-Based Modulation

In general, the OFDM-based modulation technique for optical wireless communication is depicted in Figure 1. In the transmitter, the modulator block receives a bitstream of information and proceeds to generate symbols of complex values, which is denoted as $X_k$, where $k = 0, 1, \ldots, N - 1$ for $N$ subcarriers.

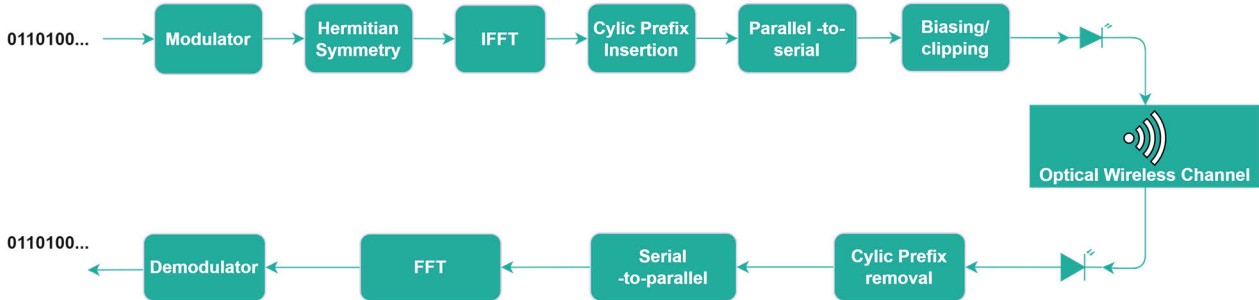

**Figure 1.** Block diagram of a system based on Orthogonal Frequency Division Multiplexing (OFDM) for optical wireless communication [31].

Furthermore, to meet the real signal requirements, the Hermitian symmetry is carried out on complex symbols with mapping as the equation:

$$X_k = X^*_{N-k} \text{ for } k = 1, 2, \ldots, N - 1$$
$$X_k = 0 \text{ for } k = 0 \text{ and } \frac{N}{2} \tag{1}$$

Inverse Fast Fourier Transform (IFFT) is then employed to derive a time-domain signal [32]:

$$x_n = \frac{1}{\sqrt{N}} \sum_{k=0}^{N-1} X_k \exp\left(j\frac{2\pi}{N}nk\right), n = 0, 1, \ldots, N - 1 \tag{2}$$

Subsequently, the cyclic prefix (CP) is inserted at the beginning of the OFDM symbol to eliminate intersymbol interference. Then, the signal is converted using a parallel to serial converter to obtain a single signal to be transmitted through the LED.

In the receiver, the light is detected by a photodetector and converted into an electrical signal. After CP removal and serial to parallel conversion, the received signal is grouped into the frequency domain by FFT using the equation:

$$R_k = \frac{1}{\sqrt{N}} \sum_{k=0}^{N-1} r_n \exp\left(-j\frac{2\pi}{N}kn\right), k = 0, 1, \ldots, N-1 \tag{3}$$

The symbols in the frequency domain are then demodulated by the demodulator block into information bits.

As shown in Figure 1, in the LiFi systems, the signal from the transmitter will pass through the optical wireless channel to get to the receiver. The indoor LiFi communication link is depicted in Figure 2, the LED is on the ceiling of the room and functions as an access point (AP) [33].

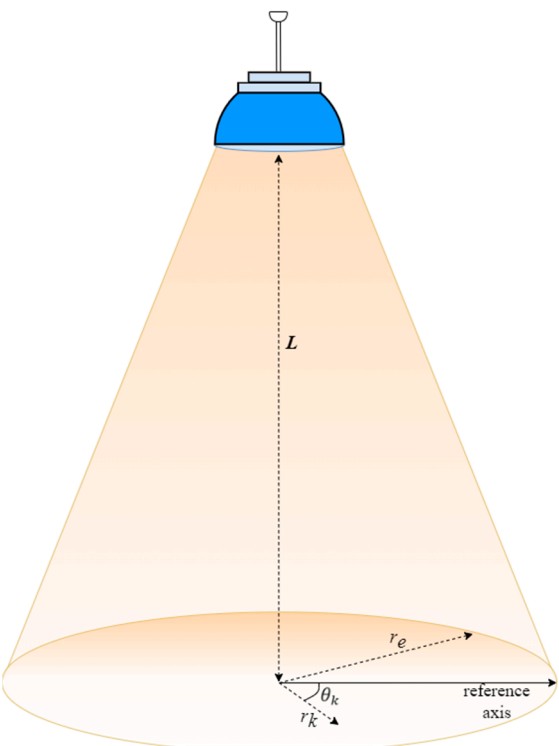

**Figure 2.** LiFi channel model [33].

The LED is at a height of $L$ from the ground, and the coverage area is circular with a radius $r_e$; this form is also called attocell. Assuming there are K users within an attocell, then the position of the $k$-th user can be denoted in the polar coordinates by $(r_k, \theta_k)$, where $r_k$ is the user's distance from the LED horizontally and $\theta_k$ is the polar angle with respect to the reference axis.

In general, the received signal can be written as:

$$Y = HS + N \tag{4}$$

where $H$ represents the channel gain, $S$ is the transmitted signal, and $N$ is the additive white Gaussian noise. Furthermore, the channel gain of LiFi is affected by the user's location and orientation of the user equipment (UE). A statistical channel modeling close to the actual condition has been carried out in [34], where the modified truncated Laplace (MTL) model and the modified Beta (MB) model have been proposed for stationary users as well as the sum of modified truncated Gaussian (SMTG) model and the sum of modified Beta (SMB) model is proposed for mobile users.

In the IM/DD system, the transmitted OFDM signal must be real and positive. Hermitian symmetry is generally chosen to obtain a real signal [21]. Meanwhile, various methods are proposed to generate a unipolar signal, resulting in many optical OFDM variants. In this paper, we adopt the classification employed in [17], which divides modulation techniques into four groups: DCO-OFDM, Inherent Unipolar OFDM, Superposition OFDM, and Hybrid OFDM.

### 2.1. DC-Biased Optical—OFDM

Direct Current-biased Optical Orthogonal Frequency Division Multiplexing (DCO-OFDM) is the most conventional technique of optical OFDM [35]. It is widely used in VLC and also LiFi due to its simplicity and high spectral efficiency [26,36]. The block diagram of DCO-OFDM is shown in Figure 3. In the transmitter, a bit stream of information is mapped into complex-valued symbols, and then, Hermitian symmetry is applied to produce a real signal.

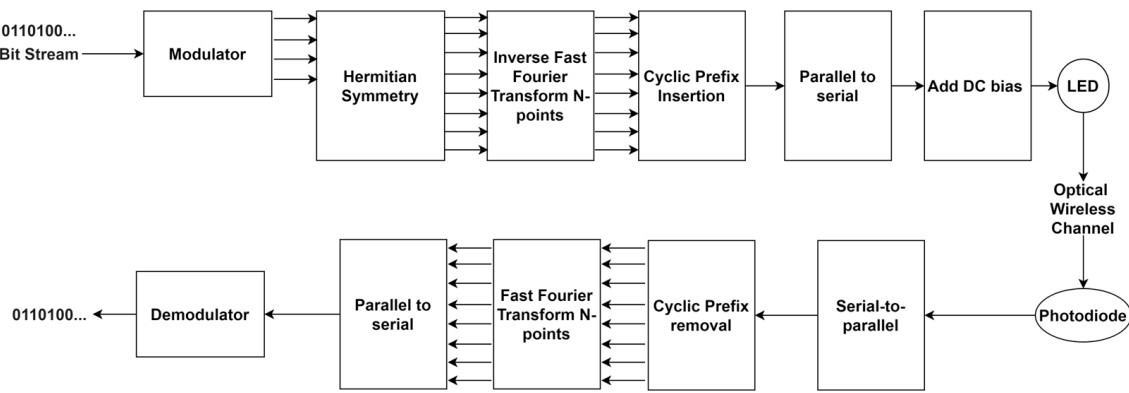

**Figure 3.** Block diagram of DCO-OFDM.

The subcarrier arrangement and Hermitian symmetry for $N$-point IFFT are depicted in Figure 4. The modulator block generates complex symbols, $S_k$ for $k = 1, 2, \ldots, N/2 - 1$. The complex symbols and those conjugates fill $N$ subcarriers while 0 th and $(N/2)$ th subcarriers are set to zero, so the frame structure becomes:

$$S = \left[ 0, S_1, S_2, \ldots, S_{\frac{N}{2}-1}, 0, S^*_{\frac{N}{2}-1}, \ldots, S^*_2, S^*_1 \right] \tag{5}$$

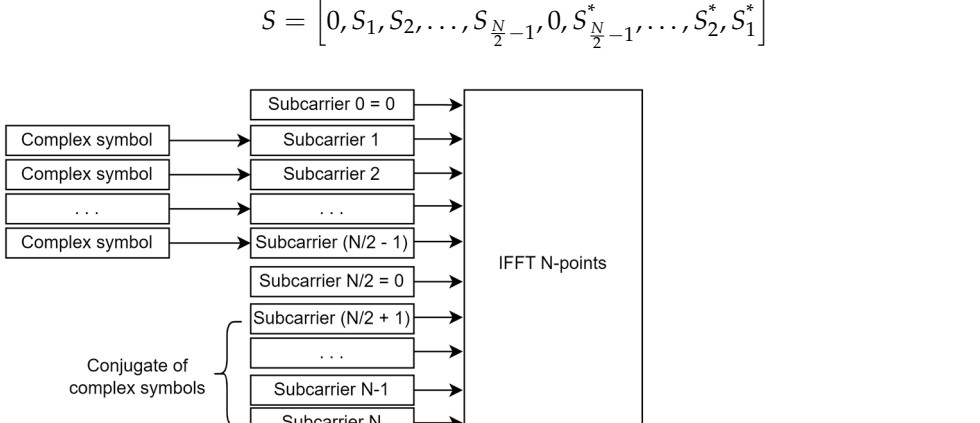

**Figure 4.** Subcarriers arrangement for $N$-points IFFT.

The Hermitian symmetry block's output is utilized as input for the $N$-sized IFFT block to obtain real-valued time-domain signals, denoted as $s_n$. These signals are further augmented with a cyclic prefix and then converted into serial format through the parallel-to-serial block.

In DCO-OFDM, the unipolar principle applied is the addition of a *DC* bias where the signal used to drive the LED $(s_{DCO})$ is:

$$s_{DCO} = s_{n,clipped} + B_{DC} \tag{6}$$

where $s_{n,clipped}$ represents the clipped signal according to the upper and lower limits, and $B_{DC}$ represents DC bias [37]. Furthermore, the reverse process is carried out on the receiver to return the bit stream of information.

## 2.2. Inherent Unipolar Optical OFDM

The Inherent Unipolar Optical OFDM group consists of Asymmetrically Clipping Optical Orthogonal Frequency Division Multiplexing (ACO-OFDM), Pulse Amplitude Modulation Discrete Multi-Tone (PAM-DMT), and Unipolar Orthogonal Frequency Division Multiplexing (U-OFDM). This group was developed to produce optical OFDM with better energy efficiency than DCO-OFDM.

The block diagram of ACO-OFDM is shown in Figure 5, the main difference from DCO-OFDM is subcarrier occupation, and there is no need to add DC bias.

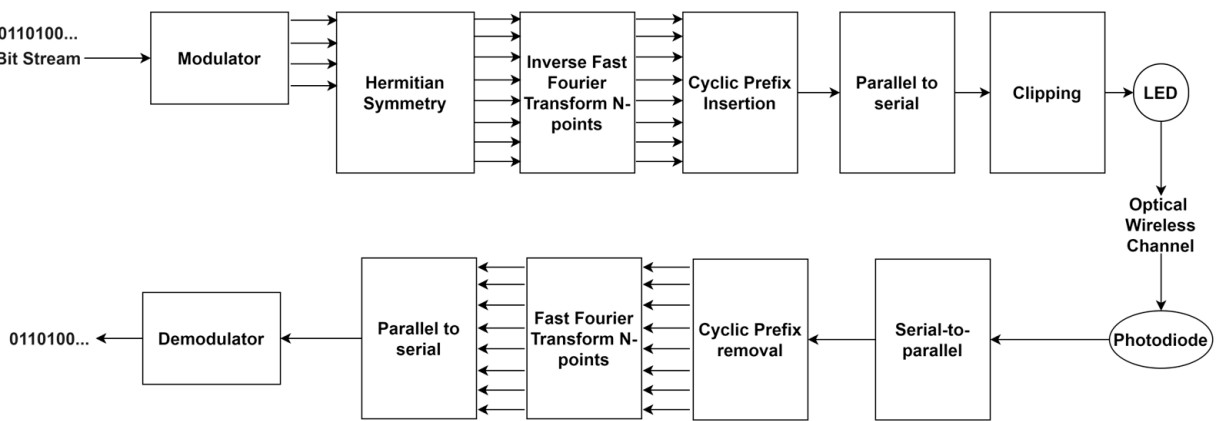

**Figure 5.** Block diagram of ACO-OFDM.

In ACO-OFDM, only odd subcarriers are used to transmit data [38] while even subcarriers are zero, so the frame structure is as follows [37]:

$$S = \left[0, S_1, 0, \ldots, S_{\frac{N}{2}-1}, 0, S^*_{\frac{N}{2}-1}, \ldots, 0, S^*_1\right] \tag{7}$$

The time-domain signal $s_n$ has anti-symmetric properties:

$$s_n = -s_{n+N/2} \text{ for } 0 \leq n < N/2 \tag{8}$$

The unipolar signal obtained by clipping the negative part at 0 (direct zero clipping) without the need for additional *DC*-bias, can be written by the equation:

$$s_{ACO} = \begin{cases} s_n, s_n \geq 0 \\ 0, s_n < 0 \end{cases} \tag{9}$$

Clipping noise only falls on the even subcarrier and does not affect the demodulation of the transmitted data.

Meanwhile, the PAM-DMT modulation technique uses the PAM symbol to modulate the imaginary part of each subcarrier, except the 0th and $N/2$th subcarriers [39]. Figure 6 illustrates the block diagram of PAM-DMT.

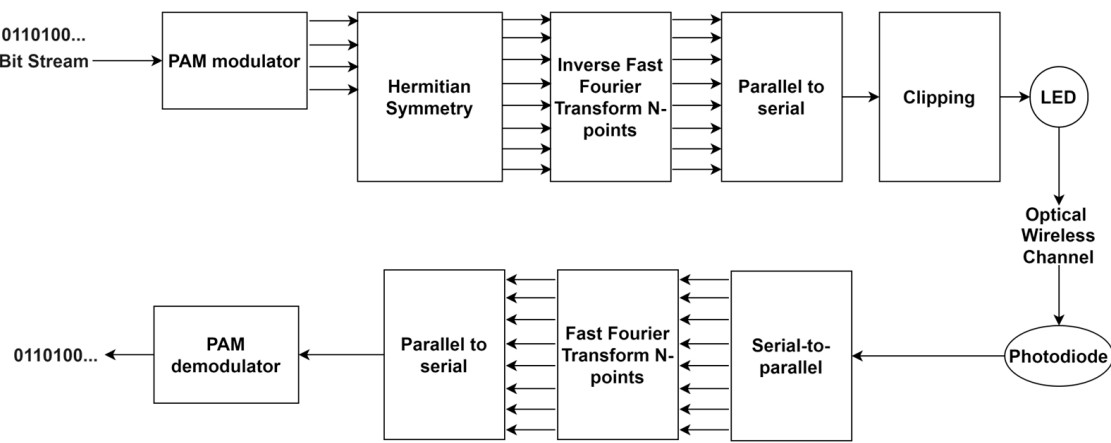

**Figure 6.** Block diagram of PAM-DMT.

The frame structure is:

$$S = \left[0, jA_1, jA_2, \dots, jA_{\frac{N}{2}-1}, 0, -jA_{\frac{N}{2}-1}, \dots, -jA_2, jA_1\right] \tag{10}$$

where $A_k$ is the real value of PAM symbols. The time-domain signal of PAM-DMT is symmetric $s_n = -s_{N-n}$ for $n = 0, \dots, N/2 - 1$ so that the signal can be clipped directly on zero as well as ACO-OFDM. Meanwhile, clipping noise only falls on the real part of the subcarrier and does not affect the data demodulation process at the receiver [37].

Furthermore, the U-OFDM modulation technique, also known as Flip-OFDM, separates the positive and negative parts of the signal, then is sent using two different frames [40]. The scheme is depicted in Figure 7.

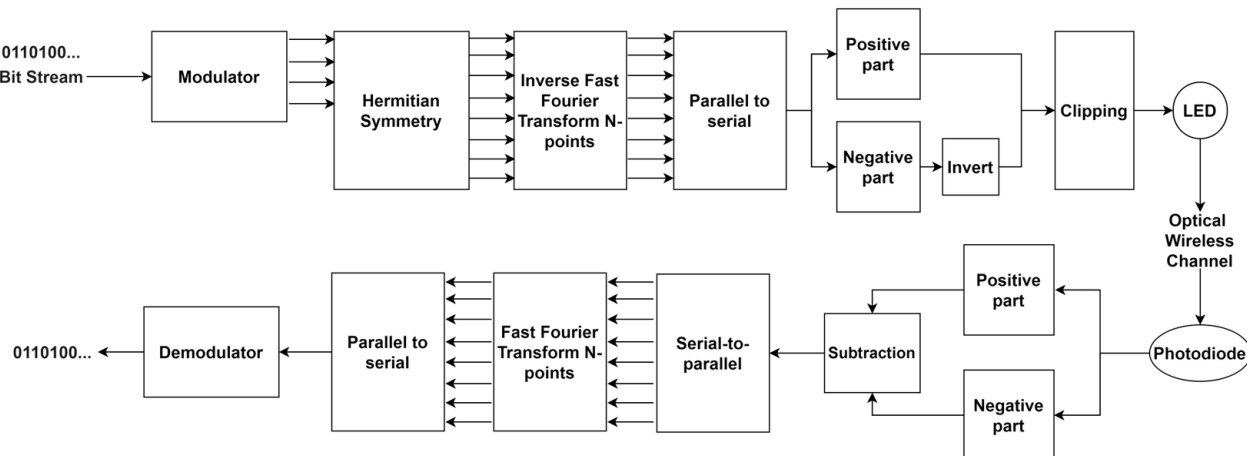

**Figure 7.** Block diagram of U-OFDM.

If $s_n$ represents the output signal of IFFT in the time domain, then $s_n = s_n^+ + s_n^-$ where:

$$s_n^+ = \begin{cases} s_n, s_n \geq 0 \\ 0, \ others \end{cases} \text{and } s_n^- = \begin{cases} s_n, s_n < 0 \\ 0, \ others \end{cases} \tag{11}$$

After that, the first frame will be sent containing the positive part $s_n^1 = s_n^+$, and the second frame contains the invert of the negative part of the signal $s_n^2 = -s_n^-$.

Since the time domain signal has been entirely positive, it is only necessary to clip the corresponding upper limit without adding a DC-bias [37]. At the receiver, two frames are

received, $y_n^1$ containing the positive part and $y_n^2$ containing the negative part. Subsequently, the time-domain signal can be obtained through subtraction:

$$y_n = y_n^1 - y_n^2 \tag{12}$$

By performing FFT on the time-domain signal and utilizing the parallel-to-serial conversion, the frequency symbols are recovered for demodulation into information bits.

### 2.3. Superposition OFDM

This group consists of modulation techniques enhanced unipolar OFDM (eU-OFDM), enhanced ACO-OFDM (eACO-OFDM), enhanced PAM-DMT (ePAM-DMT), spectrally and energy efficient (SEE-OFDM), and layered ACO-OFDM (LACO-OFDM). This group was developed on the basis that the spectral efficiency of modulation techniques of the inherent unipolar OFDM group can be increased by the superposition of several levels of OFDM waveforms.

In the eU-OFDM, the method used is to stack several U-OFDM streams by utilizing the symmetry properties of the U-OFDM frame [41]. This method has succeeded in increasing the spectral efficiency of U-OFDM, but computations with high complexity are required for signal generation and signal demodulation processes [42].

The next superposition method is to stack several ACO-OFDM streams by exploiting the symmetry properties of the ACO-OFDM subframe. This method is used in eACO-OFDM, SEE-OFDM, and LACO-OFDM. The eACO-OFDM modulation technique was developed by dividing the data stream into several depths (*d*) [43]. At each depth, the stream consists of subframes that are symmetrical in the time domain, thus facilitating demodulation at the receiver using slicing and Fourier transform techniques [44].

In the SEE-OFDM, several signals are generated, one line is ACO-OFDM for odd subcarriers and the other line is ACO-OFDM, which is processed by flip and concatenate operation to fill even subcarriers [45]. This scheme yields efficiency in terms of power and spectrum [4]. SEE OFDM only requires one Fourier transform, but noise between subcarriers arises because there is no slicing mechanism, and cyclic-prefix overhead increases due to four CPs required at once for one symbol [46].

Furthermore, the LACO-OFDM modulation technique sends data through several L layers, where each layer uses the ACO-OFDM scheme with different iteration levels. On the receiver side, demodulation is done iteratively from the first layer to the top layer. With that method, more subcarriers are used, thereby increasing the spectral efficiency [47]. Several comparisons show that LACO-OFDM achieves even better performance than the modulation technique of the Hybrid OFDM group [46,48,49].

Meanwhile, the ePAM-DMT uses the symmetry properties of the PAM-DMT subframe. Multilevel frame arrangement is made to make all the inter-stream-interference fall on the real part of the subcarrier, so it does not distort the information superimposed on the imaginary part of the subcarrier [50]. However, the performance of ePAM-DMT is not optimal due to 3 dB loss in the demodulation process at each depth, and the PAM modulation scheme is not as good as QAM [17], but we can incorporate this technique to fulfill the dimmable requirement in LiFi [51].

In general, all modulation techniques in the OFDM Superposition group achieved good spectral efficiency, which is 100% compared to DCO-OFDM. However, its implementation is influenced by latency, power penalty, computational complexity, and memory requirements, which limit the maximum depth value [17].

### 2.4. Hybrid OFDM

This group consists of Reverse Polarity Optical OFDM (RPO-OFDM), Polar OFDM (P-OFDM), Spatial Optical (SO-OFDM), Asymmetrically and Symmetrically Clipped Optical OFDM (ASCO-OFDM), Spectrally Factorized Optical OFDM (SFO-OFDM), Position Modulation (PM-OFDM), Asymmetrically clipped DC-biased Optical OFDM (ADO-OFDM), and Hybrid Asymmetrically Clipped Optical OFDM (HACO-OFDM). Basically, the hybrid

modulation technique is formed by combining several different modulation techniques, as we summarized in Table 1. In contrast to the superposition group, which uniformly provides excellent spectral efficiency, the hybrid group has different spectral efficiencies, ranging from 50% to 100% compared to DCO-OFDM. Therefore, the description in this section is limited to modulation techniques with 100% spectral efficiency only.

**Table 1.** OFDM Hybrid Modulation Techniques.

| Modulation Technique | Method | Spectral Efficiency * |
|---|---|---|
| RPO-OFDM | Real of Optical OFDM + slow PWM | 50% |
| P-OFDM | The complex signal which is the output of the IFFT is converted from Cartesian coordinates to polar coordinates | 100% |
| SO-OFDM | The subcarrier is allocated to one of the LED arrays | 100% |
| ASCO-OFDM | ACO-OFDM + SCO-OFDM | 75% |
| SFO-OFDM | Complex data goes through an auto-correlation process at the transmitter | 50% |
| PM-OFDM | The real and imaginary components of the OFDM signal are divided into positive and negative parts | 50% |
| ADO-OFDM | ACO-OFDM + DCO-OFDM | 100% |
| HACO-OFDM | ACO-OFDM + PAM-DMT | 100% |

* As a function of the spectral efficiency of DCO-OFDM.

In contrast to the commonly used OFDM technique, P-OFDM does not use hermitian symmetry to generate real signals, but the output of IFFT in the form of complex signals is converted from Cartesian coordinates to polar coordinates. The IFFT output has half-wave symmetry, so only half of it needs to be transmitted. After conversion, the radial value is sent on the first half of the OFDM symbol and the angular value of the second half. P-OFDM only uses an even subcarrier for data transmission, but the elimination of the hermitian symmetry process allows this modulation technique to achieve maximum spectral efficiency [52]. Meanwhile performance may decrease due to ISI between radial and angular samples [17].

Furthermore, SO-OFDM uses the principle of spatial summing by utilizing LED arrays. Each subcarrier is transmitted through a different LED, and when the number of LEDs increases, the PAPR decreases. SO-OFDM provides better performance than DCO-OFDM because it successfully reduces PAPR and is resistant to LED nonlinearity [53].

Meanwhile, the ADO-OFDM modulation technique uses ACO-OFDM on odd subcarriers and DCO-OFDM on even subcarriers and then transmitted simultaneously. At the receiver, the odd subcarrier will be demodulated with conventional ACO-OFDM and the even subcarrier will be demodulated after the interference cancellation process [54]. The combination of the two modulation techniques makes ADO-OFDM have good spectral efficiency [21,55], but the addition of DC bias results in energy efficiency loss [56].

HACO-OFDM uses ACO-OFDM on odd subcarriers and PAM-DMT on even subcarriers to pursue increased spectral efficiency [57]. However, at the same spectral efficiency, PAM-DMT requires a higher SNR than QAM, so only a small amount of additional data can be carried by an even subcarrier optimally; therefore, HACO-OFDM only slightly increases the spectral efficiency [46].

## 3. Performance Keys

In this section, we describe the parameters used in this paper to evaluate the performance of OFDM-based modulation techniques.

### 3.1. Energy Efficiency

The energy efficiency (*EE*) is defined as the ratio of the channel capacity over the mean power consumption [49], which is given by:

$$\eta_{EE} = \frac{C}{P} \tag{13}$$

where $C$ is the channel capacity and $P$ is the LED power. *EE* is closely related to the LiFi function as illumination [22] and is needed to realize green communication [49].

### 3.2. Spectral Efficiency

The spectral efficiency (*SE*) is the channel capacity per unit of bandwidth [49], which is given by:

$$\eta_{SE} = \frac{C}{W} \tag{14}$$

where $W$ is the bandwidth. *EE* and *SE* are important metrics in wireless communication design, but the maximum performance of these two parameters is difficult to achieve at the same time [36,49].

### 3.3. Peak-to-Average Power Ratio

The peak-to-average power ratio (PAPR) is defined as the quotient of the maximum to the average power of the time domain symbols [58]. In the IM/DD system, OFDM signal $x$ must be in real value, then the PAPR is given by:

$$PAPR = \frac{max\left\{\|x[n]\|^2\right\}}{E\left\{\|x[n]\|^2\right\}} n \in \{0, 1, \ldots, N-1\} \tag{15}$$

where $\|.\|$ is the modulo of signal, and $N$ is the number of subcarriers. The high *PAPR* is one of the major drawbacks of OFDM; it decreases the signal-to-quantization noise ratio (SQNR) of the analog-to-digital (A/D) and digital-to-analog (D/A) converters while wasting the large dynamic range of linear amplifiers [38]. Moreover, in the IM/DD system, the high *PAPR* will decrease the life span of LEDs that uses as lighting [58].

### 3.4. Computational Complexity

Computational complexity is one of the important parameters in comparing the performance of modulation techniques [59]. To explain computational complexity, we can use complexity order or commonly denote as big O notation (with capital letter O), which indicates the rate of growth of a function. For the OFDM-based modulation techniques, the complexity depends on the $N$-point IFFT/FFT operation and is expressed as $O(N\log_2(N))$ [60], which essentially takes into account the arithmetic operations of addition and multiplication.

As explained in Section 2, OFDM unipolar groups continue to be developed into superposition and hybrid OFDM groups. This development contributes to increasing the complexity of the system. For example, the ACO-OFDM, which has a receiver complexity of $O(N\log_2(N))$, when superposed to LACO-OFDM with $L$ layers, requires $L$ FFT blocks with different sizes, thus increasing the computational complexity of the receiver $(CR_L)$ to:

$$CR_L = O(N\log_2(N)) + 2\sum_{l=1}^{L} O\left(N/2^{l-1}\log_2\left(N/2^{l-1}\right)\right) \tag{16}$$

The above equation can be simplified to:

$$CR_L = \left(5 - 1/2^{L-3}\right)O(N\log_2(N)) \tag{17}$$

which indicates the complexity of the LACO-OFDM receiver is almost 5 times the complexity of ACO-OFDM, but in implementation, it is only 2 times because we can reuse the same N-point FFT/IFFT block [47].

## 4. Discussions

In this part, we will assess how well OFDM-based modulation methods fulfill the energy efficiency requirements, utilization of the frequency spectrum for transmission, PAPR, and computational complexity. We selected one technique from each category as a representative in Table 2, namely DCO-OFDM, ACO-OFDM from the Inherent Unipolar Optical OFDM group, LACO-OFDM from the Superposition OFDM group, and ADO-OFDM from the Hybrid OFDM group. We hope this table can provide a quick overview of the best modulation technique for each performance key to indicate one most feasible candidate to implement.

**Table 2.** Performance comparison.

| Parameters | Modulation Techniques | | | | Refs. |
|---|---|---|---|---|---|
| | DCO OFDM | ACO OFDM | LACO OFDM | ADO OFDM | |
| Energy efficiency | Low | High | High | Medium | [17,56,60,61] |
| Spectral efficiency | 100% | 50% | 100% | 100% | [47,48,60] |
| Peak-to-average power ratio | Best | Worst | Good | Good | [22,37,48] |
| Computational complexity | Low | Low | High | High | [47,60] |

Regarding energy efficiency, DCO-OFDM gives the lowest performance because the additional DC bias that forms a unipolar signal cannot carry information. For the same reason, ADO-OFDM also does not produce maximum efficiency in terms of optical power consumption, as it still necessitates the addition of DC bias for half of the subcarriers.

Furthermore, ACO-OFDM and other techniques from the Inherent Unipolar Optical OFDM group have succeeded in producing high energy efficiency but perform poorly in spectral efficiency. An inefficient spectrum occurs because of the data load on half the number of subcarriers. Up to this point, LACO OFDM shows the best results because it efficiently uses energy and bandwidth.

Afterward, we compare performance based on the parameter of PAPR. DCO-OFDM has the lowest PAPR value compared to other modulation techniques because the average optical power increases because of the DC-biased method used to form a unipolar signal. The same benefits of the DC-biased scheme provided by ADO-OFDM produce good PAPR. Meanwhile, LACO-OFDM also experienced an increase in average power by increasing the number of layers to yield a good PAPR value [48].

So far, LACO-OFDM is slightly superior to ADO-OFDM. However, both of these methods require a system with high computational complexity. LACO-OFDM appears to have a more complex structure than ADO-OFDM, but its complexity can be reduced using an adaptive pre-distortion method [60]. An analysis of the parameters above revealed that LACO-OFDM is the best modulation technique to implement. For this reason, we conducted further studies on the LACO-OFDM modulation technique.

LACO-OFDM was first proposed in [47] to overcome the spectral inefficiency of ACO-OFDM. Optimal results are achieved with a total of four layers, then numerical calculations show an SE of 1.875 bits/s/Hz and a data rate of 375 Mbps, thus indicating that the LACO-OFDM scheme succeeds in providing two times the spectral efficiency compared to ACO-OFDM. The subsequent issue pertains to inter-layer interference (ILI) attributable to the layered arrangement of LACO-OFDM. Therefore an improved receiver is proposed in [62], where time domain analysis is carried out on each layer followed by iterative application of a pairwise clipping technique. We summarize further research related to LACO-OFDM in Table 3, which contains the issues addressed, the proposed contributions, the methods used, and the results achieved.

**Table 3.** Comparisons of LACO-OFDM methods.

| Ref. | Authors | Issue | Contribution | Method/Result |
|------|---------|-------|--------------|---------------|
| [31] | Wang et al. (2016) | Dimming support | LACO-OFDM with a dimming control | • Set the scaling factor, direct current, and modulation order for the specified dimming level.<br>• Dimmable optical OFDM employs a multi-layer structure that can operate at varied dimming levels and achieve higher spectral efficiency than DCO-OFDM and AHO-OFDM. |
| [20] | Yang et al. (2017) | Determination of the number of layers | Adaptive LACO-OFDM with channel capacity analysis | • The ideal number of layers can be determined by simulating the maximum channel capacity and considering the bit error rate, spectral efficiency, and computational complexity.<br>• The electrical power impacts the Signal-to-Noise Ratio (SNR), whereas the SNRopt is affected by the optical power. In the case of both, a small quantity of layers suffices to achieve low SNR values, while a more significant number of layers is requisite to attain high SNR values. |
| [63] | Zhou et al. (2017) | PAPR decrement | LACO-single carrier frequency division multiplexing | • The use of the discrete Hartley transform leads to a decrease in PAPR by as much as 4.2 dB compared to traditional LACO-OFDM.<br>• The proposed system's computational complexity is less than that of conventional LACO-OFDM.<br>• LACO-SCFDM is more reliable in dealing with nonlinear transmitters and multipath fading on the channel than LACO-OFDM. |
| [64] | Song et al. (2017) | Error propagation | Pairwise coding on each layer for the LACO-OFDM scheme | • In each layer, subcarriers that have high SNR and low SNR are paired.<br>• Experiments over a 19.8 km single mode fiber transmission, 16 QAM OFDM modulation showed an appreciable reduction in BER with a slight increase in computational complexity. |
| [65] | Wang et al. (2017) | Inter-layer interference | Diversity combining for the receiver in LACO-OFDM | • By combining soft successive interference cancellation and clipping noise recovery, the novel LACO-OFDM's performance increases 2 dB in BER $10^{-6}$ than the conventional one. |

**Table 3.** *Cont.*

| Ref. | Authors | Issue | Contribution | Method/Result |
|------|---------|-------|--------------|---------------|
| [66] | Mohammed et al. (2017) | Spectral efficiency | Unificate diversity combining and layering techniques | • Clipped signal in even subcarrier is recovered and then used for diversity combining.<br>• Compared to its counterparts, the proposed LACO-OFDM receiver design provides increased performance of up to 2 dB for two layers in a flat AWGN channel. It has the best power efficiency at a high bit rate and good performance on the frequency-selective channel. |
| [58] | Zhang et al. (2017) | PAPR decrement | Tone injection method to reduce PAPR | • With a total of 200 candidate vectors for optimizing the reduction of PAPR, LACO-OFDM with three layers and 16 QAM is able to achieve a notable PAPR decrease of approximately 5 dB. |
| [67] | Wang et al. (2018) | Error propagation | Proposed two-stage receiver for LACO-OFDM | • The first stage uses soft interference cancellation (SIC) receiver with the minimum mean square error estimator.<br>• The second stage introduces two new schemes: SIC-based iterative noise clipping and SIC-based direct noise clipping.<br>• The proposed settings increase performance by 0.8 dB at BER $10^{-4}$. |
| [68] | Bai et al. (2018) | PAPR decrement | Interleaved discrete Fourier Transform spread LACO-OFDM | • Interleaved discrete Fourier Transform spread scheme consists of two methods, Hermitian symmetry-based and real and imaginary separation based. |
| [69] | Zhang et al. (2019) | Inter-layer interference | LACO-OFDM equipped with channel coding | • The forward error correction code is applied with a multi-class turbo code structure in a layered construction.<br>• With four layers, 16 QAM, and eight iterations of turbo code, the proposed scheme succeeded in overcoming ILI more efficiently than the uncoded scheme. |
| [70] | Wang et al. (2019) | Clipping noise | A decision-aided reconstruction (DAR) applied to receiver LACO-OFDM | • The proposed receiver comprises a preprocessor, DAR-based decoder, and ILI estimator.<br>• The DAR receiver can overcome clipping noise, resulting in an improvement in bit error rate performance over existing LACO-OFDM |
| [71] | Zhang et al. (2019) | Spectral efficiency | The first proposed LACO-SCFDM for a VLC system | • The suggested method achieves a 22% increase in spectrum efficiency compared to LACO-OFDM when both operate at a dimming level of 20 to 80%. |

**Table 3.** *Cont.*

| Ref. | Authors | Issue | Contribution | Method/Result |
|------|---------|-------|--------------|---------------|
| [36] | Li et al. (2019) | Inter-layer interference | Hierarchical pre-distortion method for NOMA VLC network | • The proposed scheme produces a PAPR gain over conventional LACO-OFDM by 1.68 dB for a four layers case. <br> • An experimental setup for a network of NOMA consisting of two users shows that HPD-LACO-OFDM outperforms LACO-OFDM in the bit error rate parameter. |
| [60] | Sun et al. (2019) | Inter-layer interference | Determination of pre-distortion on the layer using an adaptive scheme | • The analysis of the probability density function of the signal is used to decide which layer will undergo a pre-distortion operation. <br> • The proposed method yields better performance than non-adaptive methods, considering the tradeoff of environmental noise and computational complexity. |
| [72] | Babar et al. (2019) | Inter-layer interference | Multilayered code scheme LACO-OFDM | • Employ the forward error correction technique and successive interference cancellation to the LACO-OFDM layers. <br> • The system can operate with a capacity limit of 2 dB at BER = $10^{-4}$. |
| [73] | Li et al. (2019) | Dimming support | Reconstructed LACO-OFDM integrated with PWM | • PWM signal is set to a duty cycle of about 0.5, providing good spectral efficiency at 20 to 80% dimming levels. |
| [74] | Zhang et al. (2020) | Clipping Noise | Mathematical analysis and modeling to overcome the residual clipping noise | • Exploration of residual clipping noise in modulation techniques based on enhanced ACO-OFDM by establishing three statistical assumptions. <br> • The proposed model demonstrates the utility of a resource allocation strategy in achieving the desired symbol error rate. |
| [75] | Weiwen Hu. (2020) | PAPR decrement | Cyclic shifted LACO-OFDM | • A set of cyclic shifts is selected from the signals in the existing layers and then modulated into a complex symbol and superimposed on the odd subcarrier in the first layer. <br> • The proposed optimal optical power allocation scheme exhibits superior performance in detecting cyclic shift sets and achieving a lower average bit error rate across all layers in comparison to an equal power allocation scheme. |

**Table 3.** *Cont.*

| Ref. | Authors | Issue | Contribution | Method/Result |
|------|---------|-------|--------------|---------------|
| [76] | Zhang et al. (2020) | Channel capacity | Analysis and optimization of Discrete-Input Continuous-Output Memoryless Channel capacity on LACO-OFDM | • The findings of this study demonstrate a noteworthy capacity increase of up to 0.18 bits/symbol in 4-layer LACO-OFDM, resulting from the implementation of the suggested power-sharing strategy instead of the traditional approach. |
| [77] | Lacava et al. (2020) | Experimental | Experimental verification of multilayer channel coding | • Half-rate turbo code implemented on a system with three layers and equipped with a bit loading technique result in a 4.22 dB increase in gain for a 10 km link compared to an uncoded system. |
| [78] | Bai et al. (2020) | Spectral efficiency and power efficiency | Absolute value -LACO-OFDM | • The analysis of uncoded transmission and information rate are entailed to regulate the amount of optical power in the layers.<br>• ALACO-OFDM with fewer layers provides the best BER performance than ACO/eU/AAO/LACO—OFDM. |
| [79] | Bai et al. (2022) | Computational complexity | Low-complexity LACO-OFDM | • With half-size IFFT/FFT, computational complexity is reduced by half that of conventional LACO-OFDM and contributes to power savings.<br>• BER measurement on the line-of-sight channel and dispersive channel shows identical results with conventional LACO-OFDM |

## 5. Conclusions

In recent years, there has been a marked increase in the research and development of LiFi technology. This surge in attention can be attributed to the technology's potential to offer a significantly broader communication bandwidth than current wireless technologies. The modulation technique is one of the determinants of the data rate that can be achieved by LiFi technology. Therefore, numerous modulation schemes have been suggested by researchers, with the OFDM-based modulation approach being the most encouraging one.

The LiFi system that uses the principle of intensity modulation requires the OFDM signal to be real and positive. Therefore, it is necessary to make some modifications of OFDM to realize the IM/DD principle. This has resulted in many variants of optical OFDM for LiFi systems. Therefore, we need to find the most potential modulation technique to be implemented.

In this paper, we have reviewed OFDM-based modulation techniques and compared their performance based on the parameters of energy efficiency, spectral efficiency, PAPR value, and computational complexity. From the comparisons made, we claim that LACO-OFDM is a feasible modulation technique because it has good energy efficiency, efficient spectrum utilization, adequate PAPR value, and low computational complexity. We have further summarized the development of the method used in LACO-OFDM. The results of this review will be used for the implementation of LiFi using phosphor-coated blue LEDs in experiments on a laboratory scale.

**Author Contributions:** Conceptualization, R.A., P.S.P. and K.R.; methodology, R.A. and K.R.; software, R.A.; validation, P.S.P. and K.R.; formal analysis, P.S.P.; investigation, R.A.; resources, R.A.; data curation, R.A.; writing—original draft preparation, R.A.; writing—review and editing, R.A.; visualization, R.A.; supervision, P.S.P. and K.R.; project administration, P.S.P.; funding acquisition, P.S.P. All authors have read and agreed to the published version of the manuscript.

**Funding:** This research was funded by UNIVERSITAS INDONESIA, for "Hibah Publikasi Terindeks Internasional (PUTI)" Q2 TA 2023–2024, grant number: NKB-826/UN2.RST/HKP.05.00/2023.

**Data Availability Statement:** Not available.

**Conflicts of Interest:** The authors declare no conflict of interest.

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
