# Peer review of "Review of Orthogonal Frequency Division Multiplexing-Based Modulation Techniques for Light Fidelity"

_jlpea, doi:10.3390/jlpea13030046_

Round 1

Reviewer 1 Report

The authors have listed and compared several higher-order modulation formats on OFDM-LiFi communication systems. The topic fits the journal well and the OFDM technology is indeed essential to boost the communication capacity in terms of the limited bandwidth from existing, low-cost LEDs. The review would be useful for fellow researchers on LiFi technology to grasp the current status but also to junior researchers and students. However, I find it hard to recommend it for publication in its current form. The main problem is lack of detailed illustration and analysis of each modulation format, i.e., it seems only hand-waving introduction is added, taken directly for literature. For a comprehensive review, I suggest including detailed explanation with figures, in addition to the equation and text, and show how each modulation format works in theory and what extra steps / equipment are needed to modulate / demodulate the signals in practice. 

The scientific language should be objective, fact-based and concise. The authors need to improve the quality of their writing. E. g. I cannot interpret the claim in line 151-152 on page 4.

"In this paper, we will describe OFDM-based modulation techniques with a classification as in reference [17], wherein our opinion is quite comprehensive."

The figure caption should also be enriched to include a short introduction and explanation of the acronyms.

Reviewer 2 Report

Please refer to attached docx file

The English is very clear, except in one or two places which I have highlighted in the docx file

Round 2

Reviewer 1 Report

The authors have addressed my concerns in detail. I recommend accepting the manuscript.

The English writing has been improved.